# MAGNEx: A Model Agnostic Global Neural Explainer

## Abstract

*Black-box* decision models have been widely adopted both in industry and academia due to their excellent performance across many challenging tasks and domains. However, much criticism has been raised around modern AI systems, to a large extent due to their inability to produce *explainable* decisions that both their end-users and their developers can trust. The need for such decisions, i.e., decisions accompanied by a rationale for why they are made, has ignited much recent research. We propose MAGNEx, a global algorithm that leverages neural-network based explainers to produce rationales for any black-box decision model, neural or not. MAGNEx is model-agnostic, and thus easily generalizable across domains and applications. More importantly, MAGNEx is global, i.e., it learns to create rationales by optimizing for a number of instances at once, contrary to local methods that aim at explaining a single example. The global nature of MAGNEx has two advantages over local methods: i) it generalizes across instances hence producing more faithful explanations, ii) it is computationally more efficient during inference. Our experiments confirm that MAGNEx outperforms popular explainability algorithms both in explanation quality and in computational efficiency.

## 1 Introduction

*Black-box* decision models have for some time now posed a dilemma between power and interpretability. For use cases where explanations are necessary, the inability of black-box models to supply them is often a deterrent to adoption. However, even in low-risk scenarios this lack of explainability often causes distrust to both the developers and the users of these models, who are often puzzled about how decisions emerge (Ribeiro et al., 2016). Also, explainability is an important mechanism when investigating if black-box models act fairly and without bias (Sun et al., 2019).[1]

In this paper, we propose MAGNEx, a *model-agnostic* neural explainer that globally learns to explain an already trained model, neural or not. In this *post-hoc* interpretability setting, most methods (Ribeiro et al., 2016; Sundararajan et al., 2017; Lundberg & Lee, 2017; Luo et al., 2020) create feature-based explanations, i.e., explanations that assign a score to each feature of the input based on how important the feature is to the model's decision. This importance score may rely on some internal mechanism of the model we wish to explain, e.g., gradients (Sundararajan et al., 2017; Shrikumar et al., 2017) or attention (Jain & Wallace, 2019; Wiegreffe & Pinter, 2019). Such methods have limitations to the types of model they can explain, e.g., gradient-based methods work only with differentiable models. Perturbation-based methods, e.g., LIME (Ribeiro et al., 2016), SHAP (Lundberg & Lee, 2017), drop combinations of features for a specific input to the model and observe its output. Omitted features that have a large impact on the output of the model across perturbations are deemed important, while other features are considered unimportant. This allows perturbation-based methods to be model-agnostic, but adds severe computational complexity, since a large number of perturbations is required per input instance to create quality explanations. The search for an optimal solution by erasure (feature drop) is combinatorial and practically infeasible for even small feature spaces; the search space is the power set of the feature set resulting in a complexity of $O(2^n)$ for $n$ features. Therefore, pertubation-based methods find approximate (sub-optimal) solutions, but even this process is cumbersome, especially when the input is large (i.e., contains many features) and the explainability method is local (a different search must be performed for each input instance).

---

[1]We use *explainability* and *interpretability* interchangeably as there is no clear consensus in the literature.

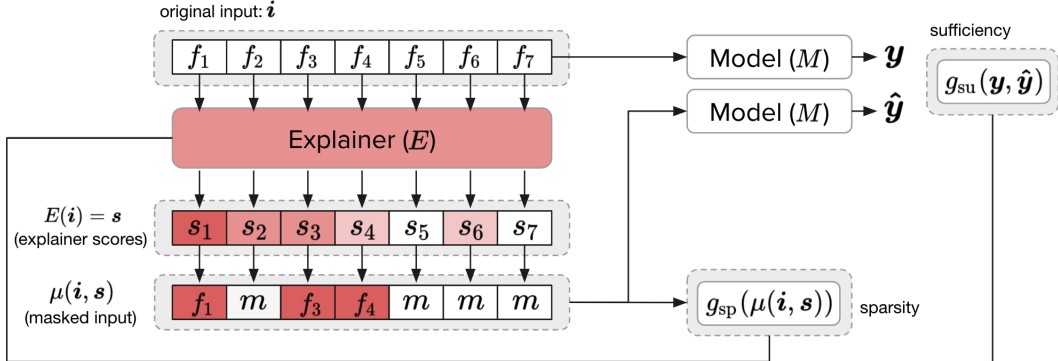

Figure 1: A high-level illustration of MAGNEx. The features of input $i$ are scored by the explainer. The scores are used to create a binary mask (Eq. 2) representing the parts of $i$ to be retained. The original and masked input are both passed through the pre-trained model and the outputs, $y$ and $\hat{y}$, are used to calculate the *sufficiency* score. The explainer aims to maximize sufficiency while also masking as many features as possible, thus also maximizing *sparsity*.

MAGNEx also creates feature-based explanations, but contrary to most methods, its explainer is a neural network that globally learns to assign feature importance scores. This is similar to the work of De Cao et al. (2020), which relies on gradients to produce feature importance scores; their explainer is a shallow network directly attached to the model being explained, requiring the gradients to flow through both the explainer and the model. This limits the approach of De Cao et al. (2020) to explaining only differentiable models, while increasing the memory and computational complexity.

Neural explanation modules have also been used in explainable *by-design* models (Lei et al., 2016; Bastings et al., 2019; Yu et al., 2019; Chang et al., 2019; Chalkidis et al., 2021). These models mainly focus on human-centric explainability and jointly train a rationale extractor and a classifier. This setting poses an extra burden on the training procedure and disincentivizes exploring parts of the space of possible solutions the model can arrive to, possibly leading to loss of model performance at the expense of interpretability. MAGNEx disentangles the explainer from the model it wishes to explain, aiming to produce explanations that are *faithful*, i.e., accurately reflect the features considered important by the model (Lipton, 2018; Jacovi & Goldberg, 2020). The generated explanations do not have to agree with human annotated rationales (e.g., gold text snippets), which are often used as an optimal solution in explainable by-design methods. MAGNEx's goal is to faithfully reveal information about a model's inner workings to its developers and if its explanations agree with human intuition, then MAGNEx can also be used as an explanation algorithm for end-users. If the explanations do not agree with human intuition, MAGNEx can reveal weaknesses of the explained model and/or of the data used to train it. For instance, training data may not be diverse enough, allowing a model to overfit to insignificant features in its input.

The explainer of MAGNEx is a shallow neural network that attributes scores to input features (Figure 1). Its training is dictated by a two-fold objective that can be classified under constrained optimization. The scoring of the explainer must be primarily *sufficient*, i.e., the explanations when fed to the underlying model must result in the same output (DeYoung et al., 2020). Also, among all possible sufficient explanations the explainer is tasked with finding one that is maximally *sparse*, i.e., utilizes the minimum number of features (Lei et al., 2016). This constrained objective, or a relaxed version of it, has been also used by perturbation-based systems (De Cao et al., 2020; Ribeiro et al., 2016).

Our contributions are as follows:

- We propose MAGNEx a method that explains the inner workings of a pre-trained model in a post-hoc manner. MAGNEx, is completely model agnostic and thus can explain any pre-trained model across modalities (e.g., vision, text) and applications.

- The proposed approach is global, allowing MAGNEx to generalize across instances; learning from many samples has a regularization effect alleviating *hindsight bias* (Fischhoff & Beyth, 1975) which is a common phenomenon in machine learning problems (Mahdavi & Rahimian, 2017) and more specificaly in perturbation-based explainability (De Cao et al.,

2020). Also, the global nature of MAGNEx transfers the computationally expensive feature search to the training stage allowing for more efficient inference than its competitors.

- The experiments showed that MAGNEx produces explanations of better quality than popular explainability methods (LIME (Ribeiro et al., 2016), IG (Sundararajan et al., 2017)), while also being more stable across instances and much more efficient during inference.

## 2 METHODOLOGY

### 2.1 FORMULATION

Let $M : I \to O$ be the model we wish to explain. Each input $i \in I$ is a set of input features such that $i = \{f_1, f_2, \ldots, f_n\}$, where $n$ is the number of features. An explainer, $E$, operates on $i$ and associates each feature in $i$ with an importance score $s_j \in [0, 1]$ for $j \in \{1, 2, \ldots, n\}$:

$$E(i) = \{s_1, s_2, \ldots, s_n\} \tag{1}$$

A masking function $\mu$, operates on an input $i = \{f_1, f_2, \ldots, f_n\}$ and its corresponding scoring $s = \{s_1, s_2, \ldots, s_n\}$ to produce a masked version of $i$:

$$\mu(i, s) = \{e_1 \in \{f_1, m\}, e_2 \in \{f_2, m\}, \ldots, e_n \in \{f_n, m\}\} \tag{2}$$

where $m$ is a feature which carries no information for $M$ and is chosen according to the task, e.g., a black pixel in computer vision or a pad token in natural language processing. In practise, $\mu$ is realized in different ways in training and inference. During training whether a feature $f_j$ will be replaced by $m$ is determined by performing a biased coin flip based on $s_j$ (the score of $f_j$), while during inference $\mu$ is realized as:

$$\mu(i, s)_j = \begin{cases} f_j, & \text{if } s_j > \epsilon \\ m, & \text{otherwise} \end{cases} \tag{3}$$

Given an input $i$ and a scoring $s = E(i)$, we compute the quality of the explainer's scoring in terms of sufficiency (su) and sparsity (sp), defined as follows:

$$su = g_{su}(M(i), M(\mu(i, s))) \tag{4}$$
$$sp = g_{sp}(\mu(i, s)) \tag{5}$$

where, $g_{su} : O \times O \to [0, 1]$ is a function measuring to what extent the output of $M$ when presented with $\mu(i, s)$ resembles the output of $M$ when presented with $i$, and $g_{sp}$ computes the percentage of features which have been replaced with $m$. Since we opt for maximally sparse inputs with high sufficiency, the total quality score for the explanation is computed as:

$$q = \begin{cases} su + sp, & \text{if } su > \upsilon \\ su, & \text{otherwise} \end{cases} \tag{6}$$

Note that we force explanations that have a sufficiency of at least $\upsilon$ to ensure a minimum quality.

Unless otherwise specified, we use a $g_{su}$ tailored to classification where we assume $O = [0, 1]^c$ and $c$ is the number of classes. Classification sufficiency is then calculated as:

$$g_{su}(\boldsymbol{y}, \hat{\boldsymbol{y}}) = 1 - (\max(\boldsymbol{y}) - \hat{\boldsymbol{y}}_p) \tag{7}$$

where $p = \text{argmax}(\boldsymbol{y})$ is the predicted class in $\boldsymbol{y} \in O$ and $g_{su}$ is bounded in $[0, 1]$ with higher scores signaling more faithful explanations. The term $\max(\boldsymbol{y}) - \hat{\boldsymbol{y}}_p$ measures the divergence between the top probability estimate across classes in $\boldsymbol{y}$ and the probability estimate for the same class in $\hat{\boldsymbol{y}}$.

### 2.2 LEARNING

Our explainer is realized as a neural network with parameters $\theta$ throughout. We aim to find the optimal values $\theta*$ which maximize q (Eq. 6) across a number of training examples $\{i_1, i_2, \ldots, i_m\} \subseteq I$. However, the standard backpropagation optimization approach falls short in this case since it is impossible to produce gradients to update the explainer. While the explainer itself is differentiable, in order for a sufficiency score to be computed for some input $i$ a hard choice must be made on

which features in $i$ to retain and which to substitute with $m$. The masking function $\mu$ is therefore non-differentiable. More importantly, our approach is completely model-agnostic and we make no assumptions about whether the model we wish to explain ($M$) is differentiable or not.

In the simplest scenario, where only the masking function is non-differentiable, a number of approaches have attempted to produce gradient estimations with methods other than the score estimator, based on REINFORCE (Williams, 1992). The most common of these approaches are the *straight-through* estimator (Chang et al., 2019; Chalkidis et al., 2021) and relaxation to binary variables (Louizos et al., 2018; Bastings et al., 2019; De Cao et al., 2020) which leverages the *reparametrization* trick (Kingma & Welling, 2013). Both of these approaches can be used only when the masking function is the only non-differentiable component, i.e., they require $M$ to be differentiable, thus breaking the *model-agnostic* nature of the explainer, which is a requirement in MAGNEX.

To retain the model-agnostic nature of our formulation and alleviate large computational strain to our method we opt to train our model by estimating gradients for updates to our explainer network with the score estimator (REINFORCE). For training on a single input $i$ we create a multi-variable policy using the output of our explainer.

$$\pi_\theta(i) = \{\mathcal{B}(E_\theta(i)_j)\}_{j=1}^{|i|} \tag{8}$$

where $\mathcal{B}$ is the Bernoulli distribution. Sampling from this policy is equivalent to sampling from each of the Bernoulli distributions independently. Therefore a sample $\tau \sim \pi_\theta(i)$ is a sequence of binary variables indicating the presence or the absence of the feature at position $j$. We train our method using the score estimator which in this case can be written as:

$$\nabla_\theta J(\pi_\theta(i)) = \nabla_\theta \mathbb{E}_{\tau \sim \pi_\theta(i)}[\mathrm{q}(\tau)] \tag{9}$$

Following Williams (1992) we can rewrite the above gradient in the form:

$$\nabla_\theta J(\pi_\theta(i)) = \mathbb{E}_{\tau \sim \pi_\theta(i)}[\nabla_\theta \log \mathrm{P}(\tau|i;\theta)\, \mathrm{q}(\tau)] \tag{10}$$

where $\mathrm{P}(\tau|\mathrm{i};\theta) = \prod_{j=1}^{|i|} P(\tau_j|\pi_\theta(i)_j)$. We can easily approximate Eq. 10 by Monte-Carlo sampling.[2] We further add a baseline in order to reduce the variance of the gradient estimator:

$$\nabla_\theta J(\pi_\theta) = \mathbb{E}_{\tau \sim \pi_\theta}[\nabla_\theta \log \mathrm{P}(\tau|i;\theta)\, (\mathrm{q}(\tau) - b)] \tag{11}$$

In Mnih et al. (2014), which bears some similarities to our setting a learned baseline was used. Here, we use a moving average baseline which seems to be sufficient for our use cases.

While similar methods relying on differentiable relaxation to binary variables have been shown to outperform REINFORCE in some cases (Bastings et al., 2019; De Cao et al., 2020), we choose this gradient estimator for a number of reasons. Firstly, we want our method to remain purely model-agnostic, a requirement which cannot be satisfied by the estimators in Bastings et al. (2019) and De Cao et al. (2020) which support only differentiable models. Secondly, the fact that the estimator works by simple exposure to a scalar metric (Eq. 6), which does not need to have a gradient, greatly reduces the space complexity of the method, allowing a higher degree of parallelism on the same hardware, and in practice allowing very complex models to be explained in reasonable time. Lastly, this problem involves constrained optimization. Looking back at the definition of our total metric (Eq. 6) we can see that we are optimizing sparsity subject to sufficiency being higher than some threshold. This objective is therefore non-differentiable and De Cao et al. (2020) and Bastings et al. (2019) employ Lagrangian relaxation to approximate the constrained objective in a differentiable manner. This adds a new hyperparameter, the Lagrangian multiplier, which needs to be tuned, adding further overhead to the explainer's development procedure and is in all cases an approximation of the true constrained objective.

## 3 EXPERIMENTS

We tested MAGNEX in three challenging settings across modalities, i.e., image classification, sentiment classification, and question answering. Although, some of these tasks are simple to handle with modern models, they pose a challenging setting for perturbation-based algorithms, since feature spaces are large compared to previous work (Ribeiro et al., 2016; Lei et al., 2016; Bastings et al., 2019; De Cao et al., 2020). This makes it difficult for such methods to identify subsets of the input which are sparse and yet expressive enough to lead to the same output.

---

[2]See Appendix A.2 for a proof.

### 3.1 Baselines and evaluation

We compare MAGNEx against Lime (Ribeiro et al., 2016) and Integrated Gradients (IG) (Sundararajan et al., 2017), two popular post-hoc explainability methods. Recall that the explainer of MAGNEx ($E$) outputs a score $s_j \in [0, 1]$ for each feature $f_j$ of an input $i$. This score can be interpreted as the probability of $f_j$ being important for a model to produce the same output. During training these probabilistic scores are converted to a binary value with a biased coin flip. During inference we cast a feature as important if its respective explainer score is above a treshold $\epsilon$ (0.5 in our experiments). On the other hand, both LIME and IG compute relative feature importance (i.e. whether a feature is more important than another) but have no explicit threshold to decide which features to keep. This is left up to the user as a post-processing step. We therefore evaluate both of these methods by selecting the top $k$ most highly scored features, where $k$ is the number of features selected by $E$ (MAGNEx) for the same input. In other words, we evaluate the sufficiency of all explanations at the sparsity level achieved by $E$. For each method we also report the time required to produce explanations averaged across inputs. Further, we intentionally make no attempt to compare explanations against human annotated rationales. Our main goal is to produce explanations which are *faithful* to the underlying model, i.e., accurately reflect the features in an input which are important for the model (Lipton, 2018; Jacovi & Goldberg, 2020), which is generally ensured when the sufficiency scores are high. Whether these faithful explanations align with what a human would consider a correct explanation is an open question and beyond the scope of this work.[3]

### 3.2 Technical details

For IG, following Sundararajan et al. (2017), we use 50 steps in the approximation of the integral throughout.[4] Concerning LIME, shallow models allow a larger number of perturbations to be drawn. Therefore, in image classification where we use MAGNEx to explain shallow models, we allow 1,500 perturbations to be drawn.[5] On the other hand, for sentiment classification the model complexity increases and we allow 1,000 perturbations, to ensure that the explanations will be created in reasonable amount of time (less than 10 seconds per input). For similar reasons, in question answering where we try to explain the most complex model in this work, we allow only 700 perturbations per example which again keeps explanation generation time around the 10 second mark per input.[6]

### 3.3 Image classification

For image classification we use the popular MNIST dataset (Deng, 2012). The input features are pixels, and we choose $E$ to be a two-layer Convolutional Neural Network (CNN) (LeCun et al., 1990) which creates a vector representation for each pixel, followed by a shared linear projection to output a single score per pixel. The feature space size is the number of pixels in an image ($28 \times 28 = 784$).[7]

#### 3.3.1 Explaining Random Forest

We train a random decision forest (Ho, 1995) to perform digit classification from raw pixels. The model achieves a perfect accuracy on the test set. In this case, we only compare against LIME because random forests are not differentiable and IG is unable to produce explanations. Table 1 shows the results. MAGNEx is able to achieve much higher sufficiency than LIME while being faster during inference. Explanations are also more stable (smaller standard deviation across inputs).

#### 3.3.2 Explaining a CNN

We also train a two-layer CNN on the same task and compare MAGNEx against LIME and IG since the CNN is able to produce gradients.[8] The CNN reaches near-perfect accuracy on the test set (99%). Our results can be seen in Table 2. All methods have very high sufficiency scores, producing faithful

---

[3]We report results for the best of two runs because the differences were small ($< 1\%$).

[4]Sundararajan et al. (2017) proposed a value between 20 and 300.

[5]We found that increasing this number further does not yield more sufficient explanations.

[6]We only perform a mild manual tuning for both baselines due to their complexity.

[7]Refer to Appendix A.3 for sample rationales.

[8]We also attempted to use a simple gradient approach (Jacobian) but the results were significantly inferior.

| Method | Sparsity | Sufficiency | Time (s) |
|---|---|---|---|
| MAGNEx (ours) | $0.81 \pm 0.05$ | $\mathbf{0.99} \pm 0.02$ | $\mathbf{0.02} \pm 0.00$ |
| LIME | – | $0.83 \pm 0.15$ | $0.30 \pm 0.02$ |

Table 1: Performance of MAGNEx and LIME when explaining a random forest model trained for digit recognition. MAGNEx outperforms LIME in both sufficiency and inference time.

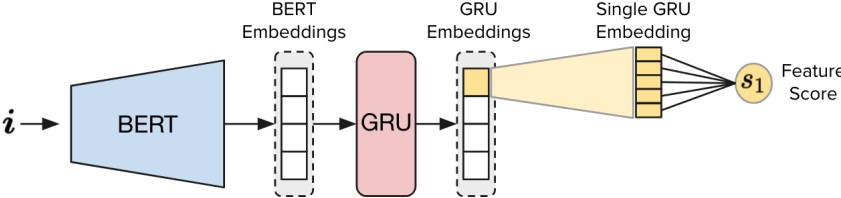

Figure 2: MAGNEx for experiments with text. The input is passed through BERT fine-tuned for the task and then a GRU further contextualizes the embeddings before each of them is passed through a shared linear layer to produce a score for each token. BERT is not updated during training.

explanations. This could be attributed to the fact that the pre-trained CNN is much more robust to slight changes in the input images. It appears to have generalized better than the random forests and can be more easily explained, as LIME performs well here while it was had difficulties when explaining the random forests. This is further supported by the fact that the sparsity here is higher, meaning that we can mask a larger percentage of each input while retaining very high sufficiency. When it comes to efficiency our method remains much faster during inference than its competitors.

| Method | Sparsity | Sufficiency | Time (s) |
|---|---|---|---|
| MAGNEx (ours) | $0.88 \pm 0.04$ | $\mathbf{0.99} \pm 0.05$ | $\mathbf{0.01} \pm 0.0$ |
| LIME | – | $\mathbf{0.99} \pm 0.06$ | $0.57 \pm 0.1$ |
| IG | – | $\mathbf{0.99} \pm 0.06$ | $0.60 \pm 0.0$ |

Table 2: Performance of MAGNEx, LIME, and IG when explaining a CNN trained for digit recognition. All methods generate sufficient explanations, but MAGNEx is more efficient during inference.

## 3.4 Sentiment Classification

We fine-tune BERT (Devlin et al., 2019) for binary sentiment classification on the IMDB review dataset (Maas et al., 2011). In this task, the input is a sequence of tokens. Again we choose a dataset with a large number of features (tokens) per instance, to ensure our method is tested for its scalability to realistic scenarios. On the validation set, the mean number of tokens per review is 303. The maximum allowed number of tokens is 512 (13% of validation instances). BERT achieves 94% accuracy on the test set. Here $E$ is a single-layer GRU model (Cho et al., 2014) operating on the contextualized embeddings drawn from the last layer of the fine-tuned BERT model. In effect the GRU further contextualizes these embeddings before passing them to a shared linear projection to create the score for each token. Updates are only performed on the GRU and the linear projection, i.e., the fine-tuned BERT remains frozen. The whole architecture of MAGNEx for this setting can be seen in Figure 2. As mentioned in Section 2.2, a reason for choosing REINFORCE is the alleviation of a lot of computational stress from the training procedure. If we were to use differentiable masking (relaxation to binary variables) to create semi-hard scores for each token, we would have to produce gradients for the feedback given back from the pre-trained model. The computation graph in this case would be further tasked with tracking gradients for the BERT model and all its intermediate computation throughout the explainer's training procedure.

Our results can be seen in Table 3. MAGNEx produces comparable or better explanations than its competitors, while being faster during inference. One additional advantage of MAGNEx is that it alleviates *hindsight bias*, which can often be observed in perturbation-based explainability. When

| Method | Sparsity | Sufficiency | Time (s) |
|--------|----------|-------------|----------|
| MAGNEX | $0.94 \pm 0.06$ | $\mathbf{0.95} \pm 0.10$ | $\mathbf{0.05} \pm 0.03$ |
| LIME | – | $\mathbf{0.95} \pm 0.09$ | $9.33 \pm 5.23$ |
| IG | – | $0.90 \pm 0.16$ | $1.32 \pm 0.70$ |

Table 3: Performance of MAGNEX, LIME, and IG, explaining BERT fine-tuned for sentiment classification. MAGNEX outperforms the baselines in both suffuciency and inference time.

this was a great movie for being only 67 minutes long . there was an aspect of film - noir contained in this movie and i am glad that nolan picked to film it in black and white . the plot is simple yet entertaining that keeps you engaged . even the dialogue was good along with the acting . it reminded me of what was to come in me ##mento by not being in chronological order . i liked how the main character tried to use what cobb taught him for example saying " everyone has a box " which he put his personal things into . also , on the writer ' s door was the batman logo which seemed ironic because christopher nolan would later direct batman begins and the dark knight , two other great movies . there is a great twist in the end which i ' m not going to spoil for anyone who hasn ' t seen it , even though i kind of figured what would happen when cobb gave the young man d lloyd ##s credit card . i also liked how the writer had a copy of the republic by plato one of my favorite philosophical books . this is definitely a movie you need to watch more than once to get the full aspect of it , plus it only being an hour long . there is also a circular aspect to it by ending where it began which i thought was pretty brilliant .

(a)

this was a great movie for being only 67 minutes long . there was an aspect of film - noir contained in this movie and i am glad that nolan picked to film it in black and white . the plot is simple yet entertaining that keeps you engaged . even the dialogue was good along with the acting . it reminded me of what was to come in me ##mento by not being in chronological order . i liked how the main character tried to use what cobb taught him for example saying " everyone has a box " which he put his personal things into . also , on the writer ' s door was the batman logo which seemed ironic because christopher nolan would later direct batman begins and the dark knight , two other great movies . there is a great twist in the end which i ' m not going to spoil for anyone who hasn ' t seen it , even though i kind of figured what would happen when cobb gave the young man d lloyd ##s credit card . i also liked how the writer had a copy of the republic by plato one of my favorite philosophical books . this is definitely a movie you need to watch more than once to get the full aspect of it , plus it only being an hour long . there is also a circular aspect to it by ending where it began which i thought was pretty brilliant .

(b)

Figure 3: Explanations produced by MAGNEX (a) and LIME (b) for a review in the IMDB test set. Both explanations have high sufficiency, but we believe that (a) is a more faithful interpretation of the model's computation. LIME appears to have fallen victim of hindsight bias identifying a single feature (*great*) as important. This does not mean that the underlying model ignores everything else. MAGNEX considers more features to be important as its global training acts against hindsight bias.

performing erasure (i.e., dropping features from the input to a model) there is no guarantee that low scored features are indeed not useful for the underlying model. The token *great* in Figure 3b is scored very highly in comparison to all other tokens. Retaining this token alone would lead to the same output. This does not necessarily mean that the pre-trained model does not consider any

other tokens when casting a decision. Although we employ no explicit mechanism to tackle this issue, we believe that we make steps towards the right direction due to two characteristics of our method. Firstly, MAGNEx is global, i.e., it learns a model across a large number of instances, which we believe to act as a regularizer against hindsight bias. The exact optimum of erasure for a specific instance can easily fall victim to *hindsight bias* and therefore approximations of this exact optimum (e.g., LIME) can often exhibit this same behaviour. On the other hand, global context allows MAGNEx to reveal patterns that generalize across instances instead of local ones. For instance, *great* being associated with a positive prediction is a pattern local to an instance that cannot be established globally. It is therefore, disincentivised as a solution in MAGNEx's global training regime. We do not argue that such patterns are non-existent, rather that they are important only when they can be established across a variety of instances and can otherwise create extremely misleading explanations (which score very well according to our metrics, i.e., they are sufficient while also being very sparse) but are at the same time not faithful explanations. Secondly, $E$ operates in the space of hidden representations, i.e., BERT token embeddings, which arguably capture contextualized morpho-syntactic and semantic information, allowing MAGNEx to learn more general explanation patterns.

### 3.5 QUESTION ANSWERING

For this task we use a large variant of BERT (24 layers, 16 attention heads, 1024 hidden size), pretrained and fine-tuned on SQUAD (Rajpurkar et al. (2016)). The model attempts to predict answer spans within some context given a question. It achieves an F1 score of $93.2\%$ and an exact match score of $86.9\%$. We show the results of various explainers on the task in Table 4. Our explainer continues to produce explanations that have higher sufficiency scores than the other methods while also being considerably faster. A side note here is that for this task sufficiency is realized as the Jaccard index. So between two predicted spans $y = \{f_1, f_2, \ldots, f_m\}$ and $\hat{y} = \{\hat{f}_1, \hat{f}_2, \ldots, \hat{f}_n\}$, sufficiency is measured as $\text{su}(y, \hat{y}) = (y \bigcup \hat{y})/(y \bigcap \hat{y})$. The Jaccard index severely punishes small mistakes. For example, the Jaccard index for the spans 'Stanford University' and 'the Standford University' is $2/3$ despite the answers being almost identical. To some extent this explains the large standard deviations (Table 4) reported for all methods. Finally, similarly to the sentiment analysis task, LIME suffers from the hindsight bias problem. Figure 4 shows an indicative example where the explanations produced by MAGNEx are more well formed than those of LIME.

| Method | Sparsity | Sufficiency | Time (s) |
|---|---|---|---|
| MAGNEx | $0.76 \pm 0.12$ | $\mathbf{0.88} \pm 0.30$ | $\mathbf{0.09} \pm 0.03$ |
| LIME | – | $0.82 \pm 0.32$ | $12.91 \pm 5.22$ |
| IG | – | $0.34 \pm 0.40$ | $5.76 \pm 2.24$ |

Table 4: Performance of MAGNEx, LIME, and IG, explaining a large variant of BERT in question answering. MAGNEx outperforms the baselines in both suffuciency and inference time.

## 4 RELATED WORK

The outburst in the field of explainability in machine learning started with Ribeiro et al. (2016) who proposed LIME, a *model-agnostic* perturbation-based algorithm that generates explanations for machine learning models. Since then, apart from perturbation-based methods, gradient based methods were explored (Shrikumar et al., 2017; Sundararajan et al., 2017)). More recently, Jain & Wallace (2019) and Wiegreffe & Pinter (2019), driven by the increasing popularity of Transformer-based models (Vaswani et al., 2017) in a number of tasks across modalities (Devlin et al., 2019; Schneider et al., 2019; Dosovitskiy et al., 2021), have initiated the discussion on whether the attention mechanism in these models can provide quality explanations.

In parallel, a lot of work focused on creating explainable neural networks *by-design* (Lei et al., 2016; Bastings et al., 2019; Yu et al., 2019; Chang et al., 2019; Chalkidis et al., 2021). These systems typically contain two components. The first component selects rationales (subsets of the input) which are then fed to the second component for classification. They are typically trained with similar objectives to perturbation-based methods, i.e., sufficiency and sparsity, other objectives such as

(a)

(b)

Figure 4: Explanations produced by MAGNEX (a) and LIME (b) for a question (green box) of SQUAD validation set. Filtering these to produce binary scores as we do during evaluation creates the reduced inputs i) *who wrote about the great pest ##ile ##nce in 1893 ? the historian francis aidan gas ##quet wrote about the ' great pest ##ile* and ii) *who wrote about ##ile ##nce 1893 ? the francis gas ##quet wrote great 1893 suggested " some adopt black interpretation other roman ##1*. Global context allows MAGNEX to produce comparatively well formed inputs to present to the pre-trained model.

*continuity* Lei et al. (2016) and *comprehensiveness* Yu et al. (2019); Chalkidis et al. (2021) have also been tested. However, the initial system (Lei et al., 2016) performed gradient estimation for the rationale extractor with REINFORCE without baseline reduction making the system unstable due to the nature of the learning algorithm. Bastings et al. (2019) relaxed this binary rationale extraction process by following Louizos et al. (2018) and therefore making the objective differentiable. Nonetheless, all further work done in this area (Yu et al., 2019; Chang et al., 2019) optimizes at least two neural networks concurrently and uses some sort of gradient estimation either through REIN-FORCE, the reparametrization trick, or straight-trough estimators. Naturally, these systems are often very hard to train due to their complexity and their innate inability to produce exact gradients due to their stochastic nodes. Finally, recent work (Jain et al., 2020; Situ et al., 2021) has attempted to use auxiliary explanations generated by some attribution method (such as LIME or integrated gradients) as supervision. For instance, one of the methods explored by Situ et al. (2021) was to train a rationale extractor on explanations produced by LIME. They also experimented with integrated gradients, sumilarly to Jain et al. (2020) to train a neural network to create explanations in the *post-hoc* setting.

## 5    CONCLUSION & FUTURE WORK

We introduced MAGNEX, a post-hoc explainability algorithm, that is global and completely model-agnostic. We experimentally showed that our approach outperforms popular post-hoc algorithms in terms of the faithfulness of its explanations and in computational complexity across tasks and modalities. In addition, the global nature of MAGNEX seems to alleviate the hindsight bias problem that seemed to trouble local perturbation-based explainability methods.

In the future we aim to pursue a number of interesting directions. We would like to combine our method with online learning to produce higher quality explanations. The explanations learned offline can be used as a starting point and further search can be performed on a per instance basis. We are also planning on performing human evaluation between MAGNEX and its baselines to determine how informative its explanations in the eyes of a human. Lastly, we will explore counterfactual explainability (Goyal et al., 2019; Elazar et al., 2021) and how it can be used to improve both the training regime of MAGNEX as well as our evaluation framework.

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

## A  APPENDIX

### A.1  IMPLEMENTATION DETAILS

We implemented all models in AllenNLP (Gardner et al., 2018) and PyTorch (Paszke et al., 2019). For experiments with text (Sections 3.4 and 3.5) we used the Transformers library of Huggingface (Wolf et al., 2020). For MAGNEx we found automatic tuning to be slow in developement since REINFORCE tends to diverge very quickly when the learning rate is non-optimal. We therefore manually tuned the hyper-parameters in each experiment by allowing a small number of batches to flow through the model and observing its metrics. Non-promising runs were terminated very quickly. We performed this procedure for sentiment classification and question answering for learning rates $\in \{10^{-4}, 10^{-5}\}$. The optimal learning rates were $10^{-4}$ for sentiment classification and $10^{-5}$ for question answering. For image classification (Section 3.3) we used a learning rate of $10^{-3}$. Optimization was carried out with Adam (Kingma & Ba, 2015) throughout the experiments except when fine-tuning BERT-base in sentiment classification where AdamW (Loshchilov & Hutter, 2019) was used. Finally, for our baselines (LIME, IG) we use Captum (Kokhlikyan et al., 2020).

### A.2  SCORE ESTIMATOR PROOF

We begin by creating a policy parametrized by $\theta$. For some input $i$ the policy can be written as:

$$\pi_\theta(\boldsymbol{i}) = \{\mathcal{B}(E_\theta(\boldsymbol{i})_j)\}_{j=1}^{|\boldsymbol{i}|} \tag{12}$$

Our loss is the expected return across samples $\boldsymbol{\tau}$ drawn from $\pi_\theta$ ($\boldsymbol{\tau} \sim \pi_\theta(\boldsymbol{i})$) with gradient:

$$\nabla_\theta J(\pi_\theta(\boldsymbol{i})) = \nabla_\theta \mathbb{E}_{\boldsymbol{\tau} \sim \pi_\theta(\boldsymbol{i})}[\text{q}(\boldsymbol{\tau})] \tag{13}$$

The probability of trajectory $\boldsymbol{\tau}$ which is a sequence of binary variables being sampled from $\pi_\theta(\boldsymbol{i})$ is:

$$\text{P}(\boldsymbol{\tau}|\boldsymbol{i};\theta) = \prod_{j=1}^{|\boldsymbol{i}|} P(\boldsymbol{\tau}_j | \pi_\theta(\boldsymbol{i})_j) \tag{14}$$

We can rewrite Eq. 13 by taking the definition of expectation as:

$$\nabla_\theta J(\pi_\theta(\boldsymbol{i})) = \nabla_\theta \int_{\boldsymbol{\tau}} \text{P}(\boldsymbol{\tau}|\boldsymbol{i};\theta)\text{q}(\tau) \tag{15}$$

We can add the gradient in Eq. 15 under the integral:

$$\nabla_\theta J(\pi_\theta(\boldsymbol{i})) = \int_{\boldsymbol{\tau}} \nabla_\theta \text{P}(\boldsymbol{\tau}|\boldsymbol{i};\theta)\text{q}(\boldsymbol{\tau}) \tag{16}$$

and rewrite Eq. 16 by utilizing $\log(f(x))' = f'(x)/f(x)$ as:

$$\nabla_\theta J(\pi_\theta(\boldsymbol{i})) = \int_{\boldsymbol{\tau}} \text{P}(\boldsymbol{\tau}|\boldsymbol{i};\theta)\nabla_\theta \log \text{P}(\boldsymbol{\tau}|\boldsymbol{i};\theta)\text{q}(\boldsymbol{\tau}) \tag{17}$$

By reverting back to expectation form we arrive at:

$$\nabla_\theta J(\pi_\theta(\boldsymbol{i})) = \mathbb{E}_{\boldsymbol{\tau} \sim \pi_\theta(\boldsymbol{i})}[\nabla_\theta \log \text{P}(\boldsymbol{\tau}|\boldsymbol{i};\theta) \ \text{q}(\boldsymbol{\tau})] \tag{18}$$

This allows us to sample $\boldsymbol{\tau} \sim \pi_\theta(\boldsymbol{i})$ and compute the gradient with Monte Carlo Sampling.

### A.3 IMAGE CLASSIFICATION RATIONALES

We show rationales for MNIST classification when the model being explained is a Random Forest (Figure 5) or a CNN (Figure 6). MAGNEX produces explanations of higher quality than LIME.

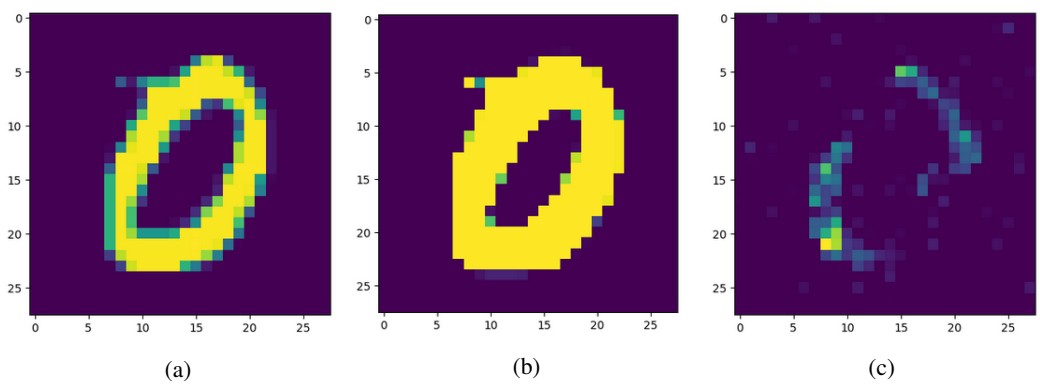

Figure 5: Explanations for a sample image (a) in the MNIST dataset with explanations generated by MAGNEX (b) and LIME (c) when explaining a Random Forest.

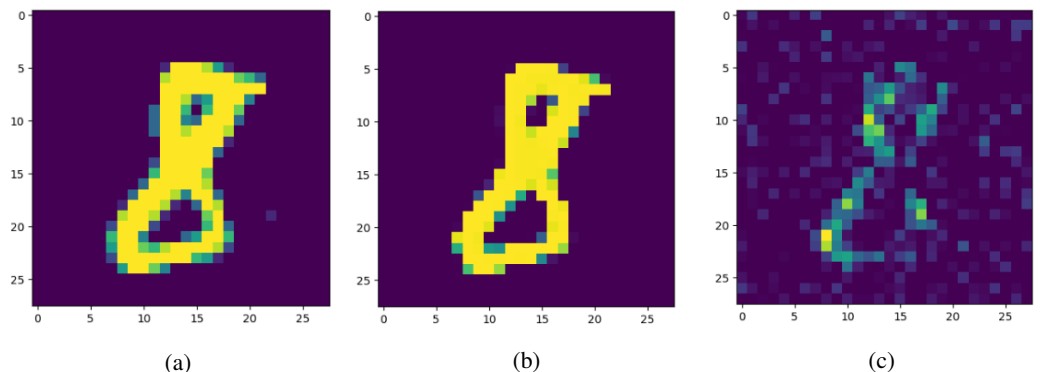

(a)               (b)               (c)

Figure 6: Explanations for a sample image (a) in the MNIST dataset with explanations generated by MAGNEX (b) and LIME (c) when explaining a CNN.

