# OpenReview forum: "MAGNEx: A Model Agnostic Global Neural Explainer"
_ICLR.cc/2022/Conference — ICLR 2022 Submitted_

### Official Review · Reviewer_eFGa · 2021-10-28

**Correctness:** 3
**Technical Novelty And Significance:** 3
**Empirical Novelty And Significance:** 3
**Recommendation:** 6
**Confidence:** 4

**Main Review:**

This is an interesting paper that discusses an important and timely topic, generating fast and reliable model-agnostic explanations. The authors propose a novel and interesting approach that compares favorably with popular methods, and present experiments across many modalities and tasks. They also discuss and analyze the methods performance, and provide a detailed review of their technical approach.

While I do find the paper useful and clear, I have some issues that I would like to see addressed for it to be published.

First and foremost, I find the lack of any discussion about causal explanation and especially counterfactuals to be highly problematic. Masking parts of the input and observing the model’s output does not causally explain the model’s decisions, even if the prediction of the model is the same.

The authors note that they intentionally do not compare to humans as we should care only about what the model is doing. However, they do not extend this reasoning to trying to causally estimate how decisions are made. I recommend the authors address the causal explanation literature, and explain why masking tokens or replacing existing pixels with black pixels is sufficient in terms of counterfactuals [1,2,3].

Second, I don’t see how is Figure 3 helpful for analysis. As I read it, this seems to be just the authors’ opinion regarding the model’s internal state, without any comparison to counterfactuals. Also, MAGNex mostly highlights “great” in this particular example, so I don’t fully agree with the intuitive explanation. Same critique stands for Figure 4 as well.

Third, the empirical results are not as convincing as the authors claim. Apart from time, MAGNEx doesn’t seem to be improving much in the first three experiments, and the fourth has huge standard deviations. In the BERT experiment, if tested on significantly shorter sentences, even the time advantage might not be true. Also, the CNN experiments should be done on a substantially more challenging dataset


[1] Explaining Classifiers with Causal Concept Effect (CaCE)
[2] Causalm: Causal model explanation through counterfactual language models
[3] Amnesic probing: Behavioral explanation with amnesic counterfactuals


**Summary Of The Paper:**

This paper proposes a global model-agnostic explanation method. The method relies on a neural model that learns to predict which input features are important for the original model’s predictions. Through substantial experimentation, the authors demonstrate that their approach outperforms LIME and Integrated Gradients. They compare between explanation methods
in terms of the faithfulness of its explanations and in computational complexity.

The main contribution of this paper is MAGNEx, a neural model-agnostic explanation method that is substantially faster than popular methods on high-dimensional inputs.


**Summary Of The Review:**

While I do find the paper interesting and find some of the contributions meaningful, I have some issues with the paper in its current form. First and foremost, I find the lack of any discussion about causal explanation and especially counterfactuals to be highly problematic. Second, the qualitative analysis of the output of MAGNEx compared with LIME is unconvincing. Third, the empirical results are not as strong as the authors claim, apart from the important and substantial gain in time efficiency.

---

> ### Author Response · Authors · 2021-11-22
> **Response to Reviewer eFGa**
>
> Thank you very much for taking the time to review our work and for the insightful comments.
> * Counterfactual explainability is indeed a very interesting direction which we plan to explore in the future. We have included this in our future work. For the time being we measured the sufficiency and the decision flips when using the complement of the mask proposed by MAGNEx for the IMDB dataset. We observed a large drop in average sufficiency (from 0.95 to 0.61) and a decision flip in 43% of the test instances. This indicates that although we do not explicitly train MAGNEx to consider counterfactual explanations it manages to some extent to extract causal explanations. Unfortunately we were unable to run similar experiments for LIME due to time limitations. Also thank you for the references we have included some in the paper.
> * In the upper part of Figure 3 we highlighted the input tokens that MAGNEx found to be important. In the lower part of Figure 3 we show highlighted the input tokens according to the importance scores returned by LIME. It is clear that MAGNEx highlights several tokens, including “great”, “glad”, “entertaining”, “definitely”, “pretty brilliant”. On the other hand, LIME has a high importance score only for the word “great”.   Similarly in Figure 4 MAGNEx highlights several words while LIME has a high importance score only for the word “who”.
> * We believe that our experimental results are quite strong. Computational complexity is an important characteristic for perturbation based explainability methods especially when large inputs are involved (which often occurs in real-world applications). Our method is consistently better on this axis throughout the experiments.  Concerning the explanations produced by the different methods,  the methods perform similarly in two experiments in terms of sufficiency, while in the others MAGNEx  has substantially better sufficiency (16% when explaining a random forest,  and 6% when explaining a question answering model) than the best performing baseline.  Even when  sufficiency is similar though,  the explanations of MAGNEx seem to be more meaningful than those of LIME (the best of the two baselines). Although we inspected several examples we only show a few due to space limitations. Concerning the complexity of our Image classification experiment we would like to explore more difficult tasks but we do not have the resources needed to experiment with datasets like ImageNet and MS COCO.

---

### Official Review · Reviewer_9rPK · 2021-10-28

**Correctness:** 3
**Technical Novelty And Significance:** 2
**Empirical Novelty And Significance:** 2
**Recommendation:** 3
**Confidence:** 4

**Main Review:**

The proposal of the paper is interesting. However, it has some weaknesses especially in the experimental part that should be fixed before publication. In particular, the evaluation is not convincing because: a) the sparsity metrics takes always the same values for all the methods tested and there are not tests of statistical significance; b) the evaluation is done with respect to two measures optimized by the MAGNEX (it would be odd that it get low results on them). However, it should be tested also with respect to other evaluation measures; c) the main advantage of MAGNEX seems to be the low runtime but this aspect is not the focus of the paper; d) there are no examples of explanations for images or question-answering; e) the approach should be tested also on the most simple data type, i.e., tabular data; f) I recommend adding a large array of baselines such as surrogate decision trees and random forests returning as explanation the features importance, SHAP, Trepan. Finally, I recommend investigating how changes the performance of these baselines when varying instead of assigned to it the same retrieve by MAGNEX.

Minor. Some mission related works:
- Setzu, Mattia, et al. "GLocalX-From Local to Global Explanations of Black Box AI Models." Artificial Intelligence 294 (2021): 103457.
- ElShawi, Radwa, et al. "ILIME: Local and Global Interpretable Model-Agnostic Explainer of Black-Box Decision." European Conference on Advances in Databases and Information Systems. Springer, Cham, 2019.
- Spinner, Thilo, et al. "explAIner: A visual analytics framework for interactive and explainable machine learning." IEEE transactions on visualization and computer graphics 26.1 (2019): 1064-1074.

**Summary Of The Paper:**

The paper proposes a global model-agnostic neural network-based explainer.

**Summary Of The Review:**

Experiments are not sound nor convincing.

---

> ### Author Response · Authors · 2021-11-22
> **Response to Reviewer 9rPK**
>
> Thank you for reviewing our work and for providing a number of comments for us to improve it.
>
> a) As we explain in the paper we consider an explanation to be good if it is sufficient (DeYoung et al., 2020) and sparse (Lei et al., 2020). In addition different inputs have by nature different sparsity levels. MAGNEx is designed to provide explanations that are sufficient and maximally sparse. On the other hand, LIME and IG only provide feature importance scores and have no mechanism that allows them to decide which features to keep; i.e. this is left to the user as a post-processing step. Therefore, we evaluate LIME and IG at the same sparsity level as MAGNex and show that at this sparsity level MAGNEx has the same or better sufficiency while also being much faster. We have updated the paper to make this clearer. With respect to statistical significance, we have not run any statistical significance tests. However, we have performed experiments with different random seeds (2 runs per experiment) and the results were very close (max. 1% difference across metrics); this has also been added in the paper.
>
> b) As mentioned in (a), sufficiency and sparsity are the two scores that characterize a good explanation. Thus, they are a reasonable selection to be used in the objective of the optimization. In general this strategy is followed in several machine learning problems. For instance in regression problems we optimize MSE and evaluate on MSE, and in classification problems we optimize cross-entropy which is the best differentiable surrogate  for Accuracy.  LIME also  optimizes for sufficiency (also known as divergence), while it implicitly optimizes for sparsity (via its Lasso regularization).  Also, optimizing for a specific metric does not guarantee that the resulting model will perform well with respect to this metric.
>
> c) One of the main features of MAGNEx is its global nature which allows for much quicker inference times than its competitors. We highlight this advantage throughout the paper. In addition, the global nature of MAGNEx allows for better generalization and more faithful explanations. This is also highlighted throughout the paper where we show that MAGNEx is comparable or better (in some cases by a large margin; from 6% to 15%). Even in the cases where the sufficiency is comparable to its competitors its explanations seem of higher quality as we show in a brief qualitative analysis.
>
> d) Figure 4 contains a question answering example. Due to lack of space we did not include computer vision examples,but we have updated the paper and have added some to the Appendix.
>
> e) MAGNEx is model agnostic and as such it can be used in nearly every setting, including tabular data. Given the space limitations we chose computer vision and NLP tasks which we believe are more challenging with respect to explainability.
>
> f) With respect to the proposed baselines, we attempted to only include baselines which could be applied to almost all of our experiments. Surrogate decision trees and random forests cannot be applied in varying size feature spaces as is the case in our NLP experiments where the model being explained is a neural network operating on sequences of arbitrary length. Using a many-hot representation to tackle the NLP downstream tasks could guarantee a fixed-size feature space (the size of the vocabulary) but such methods are outdated.  Similarly, TREPAN creates a surrogate decision tree and we excluded it from our experiments for the same reasons. We will update the manuscript to explain why we did not compare with these baselines. Regarding SHAP, the only method that could be applied in all our experiments is KernelSHAP, a model agnostic approximation for computing SHAP. However, KernelSHAP is extremely similar to LIME with some hyperparameter changes (loss function, regularization and weighting kernel) while also being extremely slow since it is a perturbation-based method. Concerning the evaluation at different sparsity levels, this is problematic for two reasons: (i)  such experiments are extremely time consuming, and (ii)  each input has a different optimal sparsity level. Thus, setting the sparsity level to the same value, e.g., 10%, across all samples will result in misleading sufficiency scores. Such an evaluation would be meaningful only if we created one curve per input example.

---

### Official Review · Reviewer_A419 · 2021-11-02

**Correctness:** 3
**Technical Novelty And Significance:** 3
**Empirical Novelty And Significance:** 3
**Recommendation:** 6
**Confidence:** 3

**Main Review:**

I think the approach in this paper is fairly intuitive. The results
are considerably better wrt execution time. One thing that was not
clear to me: is prediction time for the trimmed input going through
the original model included in the total execution time?

In the introduction, there is a statement that I'm not sure I agree
with. Last phrase of the first paragraph: "Furthermore, explainability
is an important mechanism to ensure black-box models act fairly and
without bias". I'm not sure that once we understand model predictions
we understand its fairness characteristics. Perhaps undersstanding
model predictions could help with understanding its fairness.

In Figure 4, it's not clear what the question is.

I think the paper would have a much stronger contribution if a small
sample of the results for the explainer and LIME would be judged by
humans for how helpful they are for the human to appreciate the
results of the model. As it stands now, the execution time is the main advantage (albeit, a strong one).


**Summary Of The Paper:**

This paper proposes a global explainers for black-box models. Original
inputs/features are fed in parallel with the black-box model to an
explainer model that is in charge of deciding important scores for
each feature. Depending on the scores, features are dropped and the
remaining input is also bed to the original model. The outputs for the
original input and the trimmed version with the explainer's help are
compared. The explainer is trained such that the original input and
the trimmed version produce similar scores when fed to the original
model. In addition, the sparsity of the trimmed input is maximized.

The new explainer is compared to LIME and Integrated Gradients in
terms of how faithful the produced explanations are (how close the
results of the original input and the trimmed one are when passed
through the black-box model) and the execution time on three different
tasks: image classification, sentiment analysis and question
answering. The new explainer produces the results much faster and with
similar and sometimes higher quality (as measured by faithfullness).



**Summary Of The Review:**

I think this paper introduces a straightforward technique for global explainers that has clear execution time advantages over existing explainers.

---

> ### Author Response · Authors · 2021-11-22
> **Response to Reviewer A419**
>
> Thank you for taking the time to review our work and for providing insightful comments.
> * Regarding the time reported, this is inference time; i.e, it measures how quickly an explanation method provides the trimmed version of the input. Once trained, MAGNEx does not need to pass the trimmed input through the model that is being explained. On the other hand, LIME and IG, being local methods, cannot avoid this overhead. In other words, MAGNEx transfers the overhead of using the model being explained in its training phase while LIME and IG have this overhead during inference.
> * Regarding the comment on mitigating biases and ensuring models act fairly, we think the confusion stems from us using the word “ensure”. We agree that explainability helps us understand the inner workings of the model, not “ensure” that it will act fairly. We have updated the manuscript as follows: “Furthermore, explainability is an important mechanism when investigating if black-box models act fairly and without bias”.
> * Concerning Figure 4, we have updated the paper to include a green box around the question so that it is clear where it is.
> * With respect to human evaluation, we agree that this would be very useful. Nonetheless, it requires extra resources which are not currently available. We have plans however to conduct a human evaluation in future work which we have added to our manuscript.

---

> > ### Comment · Reviewer_A419 · 2021-12-03
> > **thank you for your comments**
> >
> > Thank you for the comments provided. Good luck with your work!

---

### Official Review · Reviewer_Mi4V · 2021-11-02

**Correctness:** 3
**Technical Novelty And Significance:** 2
**Empirical Novelty And Significance:** 2
**Recommendation:** 3
**Confidence:** 4

**Details Of Ethics Concerns:**

None.

**Main Review:**

**Strengths**
1.	The proposed explainer is model-agnostic and fairly efficient, which has great potential in applications. Moreover, such universality is validated by various models, including Random Forest, BERT, and CNN.

**Weakness**
1.	I find the paper hard to read, and lots of its parts seem to be wordy and lack clarity. I suggest the authors thoroughly reorganize the lines of writing, especially for the introduction and method section.
2.	As suggested in the introduction part, the novelty of this works partly comes from the global feature of datasets instead of a single instance, which is not new in the literature, e.g., [1].
3.	Regardless of the variety of models in the experiments,  more competitive baselines are necessary to justify the superiority of MAGNEX.
4.	I think more clarifications for the experimental conclusions are needed. For example, why would the sparsity values be the same for all the methods? Plus, it would be nice to provide a more in-depth study about the performance gain of MAGNEX.


[1] Parameterized Explainer for Graph Neural Network. Dongsheng Luo, Wei Cheng, Dongkuan Xu, Wenchao Yu, Bo Zong, Haifeng Chen, Xiang Zhang.

**Summary Of The Paper:**

This paper proposes a post-hoc explainability algorithm that is global and model-agnostic. The global comes from training from batches of data while the model-agnostic property is achieved by a gradient estimator.  Experiments show the proposed method outperforms IG and LIME from aspects of sparsity, sufficiency, and time. Some case studies also validate its effectiveness.

**Summary Of The Review:**

I think this work is not ready for publication, thus I lean towards rejection.

---

> ### Author Response · Authors · 2021-11-22
> **Response to Reviewer Mi4V**
>
> Thank you for taking the time to review our work and proposing a number of improvements. With respect to the weaknesses:
> * Could you please elaborate on which parts of the introduction and the methods in paper lack clarity and need to be more well written?
> * We do not claim that MAGNEx is the first global method,  merely that it is one. Also, thank you for the missing reference. We will include it in the updated manuscript, although it is not directly comparable to MAGNEx; [1] targets only GNNs while MAGNEx is model agnostic.
> * We compare against two well known explainability methods, LIME and (IG). Could you please suggest more competitive baselines we might have missed?
> * As we explain in the paper we consider an explanation to be good if it is sufficient (DeYoung et al., 2020) and sparse (Lei et al., 2020). In addition, different inputs have by nature different sparsity levels. MAGNEx is designed to provide explanations that are sufficient and maximally sparse. On the other hand, LIME and IG only provide feature importance scores and have no mechanism that allows them to decide which features to keep; i.e. this is left to the user as a post-processing step. Therefore, we evaluate LIME and IG at the same sparsity level as MAGNex and show that at this sparsity level MAGNEx has the same or better sufficiency while also being much faster. We have updated the manuscript to better explain the reason for this evaluation scheme.

---

### Decision · Program_Chairs · 2022-01-20

**Decision:**

Reject

**Comment:**

This paper proposes a global model-agnostic explanation method. The method relies on a neural model that learns to predict which input features are important for the original model’s predictions. Using experimentation, the authors demonstrate that their approach outperforms LIME and Integrated Gradients. They compare the explanation methods in terms of the faithfulness of explanations and computational complexity. While the premise of the work is interesting, reviewers have suggested several areas for improvement: i) writing can use more clarity. writing seems verbose and hard to understand especially in intro and methods section ii) several points have been raised about empirical evaluation including lack of user studies, a more extensive set of baselines and data modalities. Given this, we are unable to recommend an acceptance at this time. We hope the authors find the reviews helpful.